# Adapted "Break the Cycle for Avant Garde" intervention to reduce injection assisting and promoting behaviours in people who inject drugs in Tallinn, Estonia: A pre- post trial

Anneli Uusküla[1]*, Mait Raag[1], David M. Barnes[2‡], Susan Tross[3‡], Talu Ave[1‡], Don C. Des Jarlais[2]

1 Department of Family medicine and Public Health, University of Tartu, Tartu, Estonia, 2 College of Global Public Health, New York University, New York, New York, United States of America, 3 HIV Center for Clinical and Behavioural Studies, Department of Psychiatry, Columbia University Medical Center, New York, New York, United States of America

☯ These authors contributed equally to this work.
‡ DMB, ST and TA also contributed equally to this work.
* anneli.uuskula@ut.ee

**Data Availability Statement:** The data are not publicly available. The data contain sensitive patient information and sharing restrictions are imposed

## Abstract

In the context of established and emerging injection drug use epidemics, there is a need to prevent and avert injection drug use. We tested the hypothesis that an individual motivation and skills building counselling, adapted and enhanced from Hunt's Break the Cycle intervention targeting persons currently injecting drugs would lead to reduction in injection initiation-related behaviours among PWID in Tallinn, Estonia. For this quasi-experimental study, pre-post outcome measures included self-reported promoting behaviours (speaking positively about injecting to non-injectors, injecting in front of non-injectors, offering to give a first injection) and injection initiation behaviours (assisting with or giving a first injection) during the previous 6 months. Of 214 PWID recruited, 189 were retained (88.3%) for the follow-up at 6 months. The proportion of those who had injected in front of non-PWID significantly declined from 15.9% to 8.5%, and reporting assisting with 1st injection from 6.4% to 1.06%. Of the current injectors retained in the study, 17.5% reported not injecting drugs at the follow up. The intervention adapted for the use in the setting of high prevalence of HIV and relatively low prevalence of injection assisting, tested proved to be effective and safe.

## Introduction

Injection drug use (IDU) is an important driver of the HIV epidemic worldwide. It is also a significant source of other morbidity (non-lethal overdose, attempted suicide, skin and soft tissue infections) and mortality [1, 2]. According to a global review of injection drug use and HIV epidemiology at a regional level, prevalence of injection drug use varied from 0.09% (95% UI 0.07–0.11) in South Asia to 1.30% (0.71–2.15) in Eastern Europe [3]. According to a recent review, four countries (Russia, Brazil, China, and the United States) comprised 55% of the estimated global population of PWID living with HIV).

by the Ethics Review Board of the University of Tartu. Data requests might be sent to the secretary of the Research Ethics Committee at eetikakomitee@ut.ee.

**Funding:** This work was supported by grant 1DP1DA039542 from the National Institutes of Health, USA, and by grant # IUT34-17 from the Estonian Ministry of Education and Research.

**Competing interests:** The authors have declared that no competing interests exist.

However, injection drug use driven HIV epidemics are also occurring elsewhere. Over the last decade, HIV outbreaks occurred among PWID in Canada (southeastern Saskatchewan), Greece (Athens), Ireland (Dublin), Israel (Tel Aviv), Luxembourg, Romania (Bucharest), Scotland (Glasgow), [4] and USA [5].

It has been suggested that the most effective method of preventing injection-driven HIV epidemics was to shift resources upstream, towards the prevention of injection drug use itself [6]. Injecting illicit drugs is a complicated, potentially fatal process. Almost every person who injects drugs needs the assistance of an experienced PWID with their first injection (initiation). The existing knowledge base on strategies to prevent injecting initiation is limited. Among a small number of studies, behavioural interventions, delivered by counsellors or peers in recovery, have been found to be effective [7–10]. Studies suggest that the majority of injection initiation events are facilitated, either directly or indirectly, by experienced PWID [9]. Although it is possible to learn to inject without the help of a PWID, this is difficult and rare [11, 12].

Social cognitive theory [13] is a useful paradigm for understanding this. Social cognitive theory hypothesizes that people learn and modify behaviours through interaction, observation, behavioural experimentation, and reinforcement with others in their environments. Repeated exposure, either through verbal or visual modelling of a marginal or even feared behaviour, can make the behaviour seem normal and acceptable by desensitizing the observer to the possible risks of the behaviour. According to social cognitive theory, three fundamental processes could drive initiation of injection. These are: (1) social modelling of injection, and concomitant interest in emulating one's injecting friends; (2) development of outcome expectancies about injection—including both enhanced positive expectancies (e.g. that injecting will produce a more intense, more efficient, cheaper high) and decreased negative expectancies (e.g. that injecting will produce stronger need and greater harms to health and life); and (3) development of self-efficacy about injecting on one's own.

Based on the theory and knowledge from behavioural interventions with experienced PWID to reduce injection initiation in non-injectors [7, 9, 14, 15] we have proposed a multistage model of how a PWID comes to assist persons who do not inject with their first injections [16]. It is reasoned that interventions that equip experienced PWID with the skills and motivation to limit behaviours that help initiation can reduce initiation. There are two types of such behaviours: (1) "assisting" behaviours—including describing or demonstrating how to inject to a non-PWID, or actually injecting a non-PWID) and (2) "promoting" behaviours—including speaking positively about injecting to non-PWIDs, injecting in front of non-PWIDs, and offering to give a first injection to non-PWIDs) [16]. In particular, the "Break The Cycle" intervention, for coaching PWID to refrain from these behaviours, has been shown to be effective in reducing these behaviours. Originally conducted by counsellors or outreach workers [7, 15], it has also since been conducted, as "Change The Cycle", by PWID themselves [9].

The objective of the current study is to assess changes to injection initiation assisting and promoting behaviours in participating PWID, during the six months following an adapted BTC session, using motivational interviewing and behavioural skills training.

We have updated/adapted BTC for use in an Eastern European setting (Tallinn, Estonia). Estonia experienced a very large epidemic of injecting drug use beginning in the 1990s and a very high seroprevalence epidemic of HIV ($> 50\%$ prevalence) among PWID since in the 2000s [17], with new injectors continuously exhibiting high-risk behavior and correspondingly high HIV prevalence also in the recent studies [18]. Community Needle and Syringe Programs (NSP), methadone maintenance treatment, and naloxone distribution programs were operating in Tallinn at the time of the study execution. The proportion of PWID receiving ART in Tallinn has increased substantially over the years, reaching over 70% among HIV-infected PWID [17].

## Methods

### Study design

We report data from a quasi-experimental study with pre-post design was used to assess potential change in assisting and promoting behaviours from baseline to 6-month follow-up in participating PWID. A standardized study protocol was implemented. The adaption process and study protocol are described in detail elsewhere [10].

### Study setting and participants

From December 2018 to April 2019 current PWID recruited by respondent driven sampling in Tallinn were enrolled. The NSP of NGO Convictus (fixed site) was the study site, given that: (1) It has established contacts and working experience with PWID; (2) It is providing HIV prevention services to and is trusted by the PWID community; (3) The site leader and staff have a track record of conducting research, including participation in international research teams, and have undergone extensive training in the conduct of scientific research.

Potential participants were eligible for the study if they: live in Tallinn or Harju County, were at least 18 years of age, spoke Estonian or Russian, reported having injected in the previous two months, and were able and willing to provide informed consent and agreed to donate a blood sample for HIV testing.

Recruitment began with purposive selection of "seeds" (n = 8) known to the field team to represent PWID diverse by age, gender, ethnicity, main type of drug used, and HIV status, and length of injecting career. After study participation, subjects were provided coupons for recruiting up to three peers (PWID). Coupons were uniquely coded to link participants to their survey responses and to biological specimens, and for monitoring recruitment lineages. Participants received a primary incentive (a 10-euro grocery store voucher) for their time and effort and a secondary incentive (a 5-euro grocery store voucher) for each peer recruited. Peers had to come to the study site, be found eligible, and complete the study procedures for the recruiter to receive the secondary incentive.

### Study procedures

After determining eligibility and securing informed consent, participants completed a face-to-face interviewer administered structured questionnaire of approximately 30–45 minutes' length in a private location in the NSP.

Venous blood was collected from participants and tested for the presence of HIV antibodies using commercially available test kits (ADVIA Centaur CHIV Ag/Ab Combo [SIEMENS]). Participants received pre- and post-HIV test counselling.

The intervention, that on average took 40 minutes, was delivered after blood collection.

At six months' post-baseline, the research coordinator reminded participants about their follow-up visit by phone, text message or email (according to participants' preference). At the follow-up visit, the data were collected in the same way as at baseline, and participants received a supermarket voucher with a 10-euro grocery store voucher for their time and effort.

Study data were managed anonymously, based on codes assigned to the participants for the study purposes.

### Measuring injection initiation assisting and promoting behaviours and background variables

The same interview-administered structured questionnaire was used throughout the study used an interview-administered structured questionnaire. The instrument used contains

multiple choice answer options and rating scales, and is based on the WHO Drug Injecting Study Phase II survey [19].

Outcome variables: Our primary outcome was assisting with a first injection. "Assisting" consisted of describing or demonstrating how to inject to a non-PWID who then injects for his/her first time in front of the participant, or actually injecting a non-PWID. Participants were asked about number of non-PWID they had assisted in the past six months. The secondary outcome was promoting the first injection. "Promoting" that consisted of speaking positively about injecting to non-PWIDs, injecting in front of non-PWIDs, and offering to give a first injection to non-PWIDs. Participants were asked about number of non-PWID with whom they had promoted injection in the past six months. We note that assisting behaviours are distinct from promoting behaviours. Whereas the former by definition (see above) intentionally lead directly to someone's first injection, promoting behaviours may or may not lead to someone's first injection.

Background variables: Questions also elicited information on PWIDs' demographics, injection and other drug use, sexual risk behaviour, HIV- and addiction- related stigma, psychological and physical health, and use of various HIV/harm reduction-related services. Other questions elicited information on size of PWIDs' injecting and non-injecting drug using peer networks (using standard RDS network questions [20]). To assess injection initiation helping peer norms, we asked participants to estimate the proportion of their PWID peers who have assisted with first injections in the last six months.

## Sample size calculation

We assumed the proportion of those who start [assisting / any promoting] to be at most 20% and the proportion of those who stop [assisting / any promoting] to be at least 60% [10]. To achieve at least 80% power using 1-sided sign test the sample size needed was 160.

## Intervention—Break the Cycle for Avant Garde (BtCag)

The intervention consisted of one individual session with a trained interventionist (social worker, psychologists, and harm reduction workers who were experienced in working with people who use drugs). They participated in two-day intervention training led by two clinical psychologists with extensive experience in motivational interviewing with drug using populations—combining didactic information, skill modelling, role playing, and feedback. At the end of training, the trainers assessed mock sessions for fidelity to the intervention. All interventionists were found to have demonstrated fidelity. Most interventionists also had formal training in Motivational Interviewing (MI) prior to the study.

The centrepiece of the intervention was the "Break the Cycle" (BTC) intervention [7] aimed at enhancing current injectors' motivation and skills to avoid helping non-injecting drug users transition to injection drug use. It was based on two conceptualizations of behavior to behaviour change. One component was Social Cognitive Theory—which, as described earlier, explains behaviour change as the result of peer modelling, expectancies about the target behaviour, and perceived self-efficacy (to carry out the target behaviour). The second component was Motivational Interviewing [21]. It is a client-centred approach—which seeks to meet the individual where he/she is in the process of behaviour change. Because such behaviour change presents both positives and negatives for the individual, MI proceeds from the premise that ambivalence about behaviour change presses for action. MI is a process aimed at articulating that ambivalence, assessing positives and negatives and the disparity between them, and pinpointing a next action step.

Our enhanced BtCag intervention had seven main parts: 1) discussion of own first time injecting drugs; 2) discussion of injection helping ("assisting" and "promoting") behaviours, experiences with and attitudes toward them; 3) discussion of the health, legal, social, and emotional risks of injection (including a module on safe injection practices); 4) role-plays of behaviours and scripts for avoiding or refusing requests to help non-PWID inject for the first time; 5) role-plays of talking with other PWID about not encouraging non-PWID to start injecting; 6) discussion of coaching non-PWID in safer injection practices, should a helping situation take place; and 7) discussion of how naloxone can be used to reverse overdoses. Guided by prior qualitative interviews with PWID, we augmented the original BTC with parts 5, 6, and 7 [22].

Intervention fidelity was maintained through audio recording and review of 10% of intervention sessions. In-group supervision meetings, the supervisor and team provided feedback, practical advice and support to the interventionists.

### Statistical analysis

We used statistical environment R [23] for analyses. Compared to RDS sequential-sampling-weighted estimates, [24] the unweighted estimates did not vary significantly from the weighted estimates for our key variables, e.g., demographics, drug use behaviours, assisting others with a first injection, and injection promoting behaviours. We therefore used the unweighted values in order to facilitate comparisons with other Break the Cycle studies that did not use RDS recruitment.

Factors associated with loss to follow-up were examined using multivariable logistic regression (using backwards elimination), from which adjusted ORs with corresponding 95% CIs, were estimated. Factors significantly associated with the loss to follow-up at an α level of 0.2 in a bivariable analysis were included in the multivariable model.

For the main analysis, the null hypothesis was, that participation in Break the Cycle will not change the percentage of participants reporting "assisting" behaviours in the 6 months prior to intervention compared to the 6 months post intervention. We tested the hypotheses that participation in Break the Cycle will be associated with a decline, from six months prior to baseline to six months' post-intervention, in: (1) percentage of participants reporting "assisting" behaviours: and (2) percentage reporting "promoting" behaviours. Based on previous research [3, 5], showing strong findings of such declines, we used one-tailed hypothesis testing. In addition, given the international importance of developing interventions to reduce initiation into injecting drug use, we believed it to be crucial to avoid type II error. We used the sign test to assess the probability of reduction in target behaviours compared to the probability of increase in target behaviours.

Ethical approvals for the studies were obtained from the Ethics Review Board of the University of Tartu, Estonia and from the Mount Sinai Beth Israel Medical Center, and New York University Institutional Review Boards in New York, USA (i.e., the home institution of the US collaborators). Written informed consent was obtained from all participants. The study is registered at the ClinicalTrials.gov (NCT03502525).

## Results

### Study sample characteristics

Study subjects' recruitment and retention are shown in Fig 1. The demographic, drug and injection use and HIV characteristics of sample recruited (n = 214) are presented in Table 1. The mean age of the sample was 35,9 (SD 7,0; sample median 35) years, ranging from 21 to 60 years. Over two thirds (71.0%) of the PWID were men, and half had 10 or more years of formal

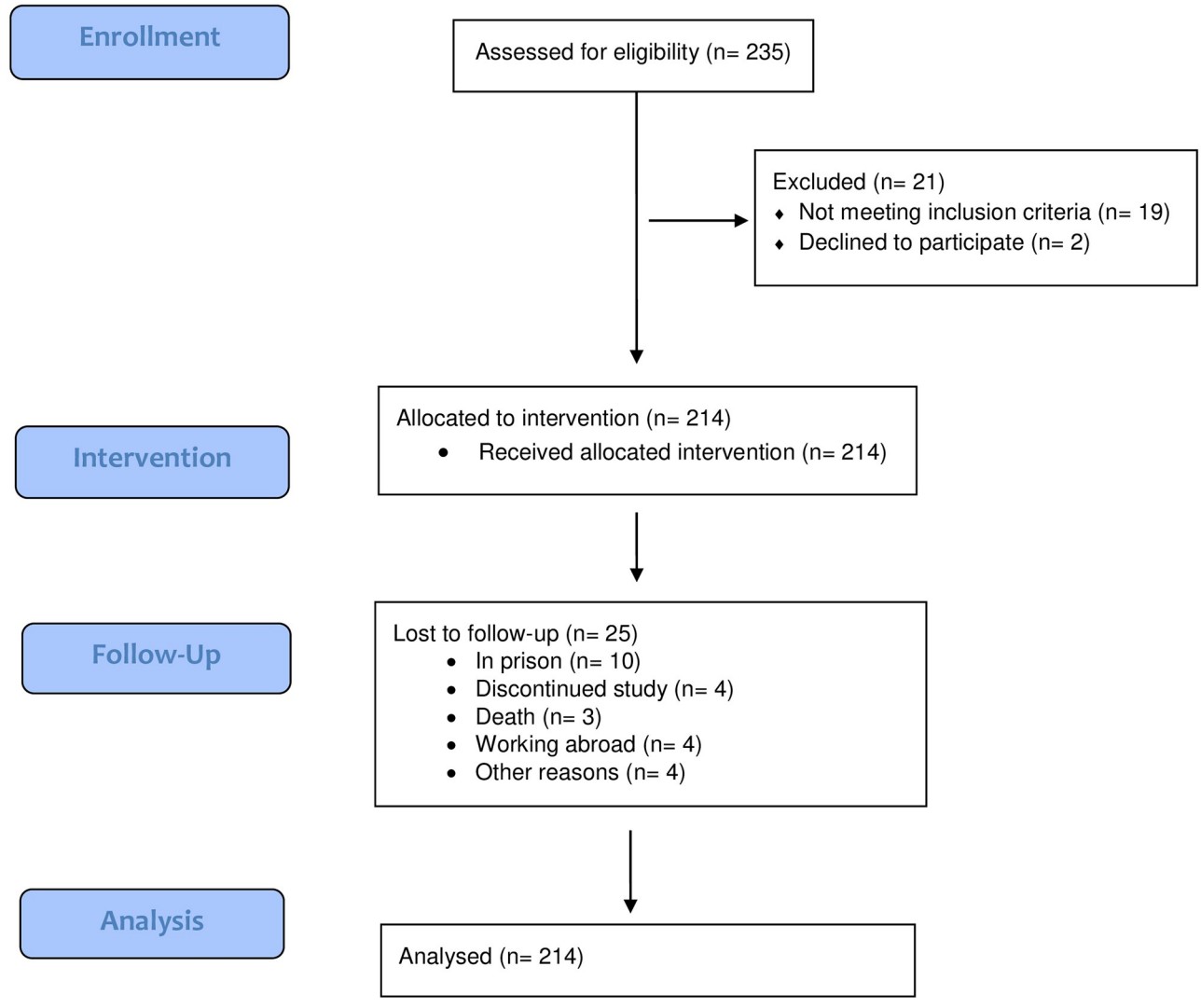

**Fig 1. Flowchart of the study.**

education and were employed (50.9%%), and 17.3% had unstable places of residence (e.g., they lived primarily in the street, a park, or in a shelter). An overwhelming majority (94.4%) had injected drugs for over five years, and reported non-injection drug use in parallel with injecting (94.9%). Amphetamine was a main injection drug for 61.8%, and fentanyl for 37.4% of PWID participating. Receptive and distributive sharing of syringes and needles (i.e., injecting with used syringes and give used syringes to others to inject with) over the past six months were, respectively, reported by 16.9% and 19.2%. Over half (51.4%) of participants were HIV infected, and a large majority (93.0%) were seropositive for HCV antibodies. Of those who were HIV seropositive, 73.6% were on ART.

Of the 214 people who received the intervention, 189 were retained (88.3%). Attrition was associated with sex (being male; aOR 13.0, 95% CI 2.1–64.4, p = 0,0234), and the main source of new syringes (other than NSP; aOR aOR 10.0, 95% CI 2.6–47.2, p = 0,0016). Attrition was not associated with injection promotion (aOR 1.7, 95% CI 0.1–2.7, p = 0.4710) or assisting behaviors (OR 1.9, 95% CI 0.1–14.5, p = 0.5901) reported at baseline.

**Table 1. Sample description (baseline and follow-up at 6 months), people who inject drugs, Tallinn, Estonia 2018–2019.**

| Variable | Categories | Baseline n (%) | | | Retained vs lost to follow-up |
|---|---|---|---|---|---|
| | | All | PWID retained | PWID lost to follow up | p-value |
| | | N = 214 | n = 189 | n = 25 | |
| Age | > 30 | 161 (75,2%) | 145 (76,7%) | 16 (64,0%) | 0,1710 |
| | < = 30 | 53 (27,8%) | 44 (23,3%) | 9 (36,0%) | |
| Gender | Female | 62 (29,0%) | 61 (32,2%) | 1 (4,0%) | 0,0035 |
| | Male | 152 (71,0%) | 128 (67,7%) | 24 (96,0%) | |
| Education | <10 years | 107 (50,0%) | 95 (50,3%) | 12 (48,0%) | 0,8315 |
| | > = 10 years | 107 (50,0%) | 94 (49,7%) | 13 (52,0%) | |
| Employment | Not employed | 105 (49,1%) | 89 (47,1%) | 16 (64,0%) | 0,1170 |
| | Employed | 109 (50,9%) | 100 (52,9%) | 9 (36,0%) | |
| Place of residence | Unstable housing | 37 (17,3%) | 32 (16,9%) | 5 (20,0%) | 0,7034 |
| | Stable housing | 177 (82,7%) | 157 (83,1%) | 20 (80,0%) | |
| **Injection drug use (in the last 6 months)** | | | | | |
| Length of injection drug use (lifetime) | < = 5 years | 12 (5,6%) | 8 (4,2%) | 4 (16,0%) | 0,0159 |
| | > 5 years | 202 (94,4%) | 181 (95,8%) | 21 (84,0%) | |
| Main drug injected | Other | 134 (62,6%) | 118 (62,4%) | 16 (64%) | 0,8790 |
| | Fentanyl | 80 (37,4%) | 71 (37,6%) | 9 36%) | |
| Any non-injection drug use | No | 11 (5,1%) | 11 (5,8%) | 0 (0%) | 0,9896 |
| | Yes | 203 (94,9%) | 178 (94,2%) | 25 (100%) | |
| Injecting daily (in the last 4 weeks) | Daily | 44 (23,5%) | 37 (21,6%) | 7 (43,8%) | 0,0538 |
| | Less frequent | 143 (76,5%) | 134 (78,4%) | 9 (56,2%) | |
| Receptive sharing [1] | No | 177 (83,1%) | 156 (83,0%) | 21 (84,0%) | 0,8980 |
| | Yes | 36 (16,9%) | 32 (17,0%) | 4 (16,0%) | |
| Distributive sharing [2] | No | 172 (80,1%) | 153 (81,4%) | 19 (76,0%) | 0,0523 |
| | Yes | 41 (19,2%) | 35 (18,6%) | 6 (24,0%) | |
| **Sexual behaviour (in the last 6 months)** | | | | | |
| Any sex partners | Yes | 182 (85,4%) | 163 (86,2%) | 19 (79,2%) | 0,3584 |
| | No | 31 (14,4%) | 26 (13,8%) | 6 (20,8%) | |
| ..Any unprotected sex | Yes | 149 (81,9%) | 136 (83,4%) | 13 (68,4%) | 0,1160 |
| | No | 33 (18,1%) | 27 (16,6%) | 6 (31,6%) | |
| **HIV infection** | | | | | |
| HIV seropositivity | Pos | 110 (51,4%) | 103 (54,5%) | 7 (28,0%) | 0,0129 |
| | Neg | 104 (49,6%) | 86 (44,5%) | 18 (72,0%) | |
| **Services utilization** | | | | | |

*(Continued)*

**Table 1.** (Continued)

| Variable | Categories | Baseline n (%) | | | Retained vs lost to follow-up |
|---|---|---|---|---|---|
| | | All | PWID retained | PWID lost to follow up | p-value |
| | | N = 214 | n = 189 | n = 25 | |
| Currently on methadone | No | 202 (94,4%) | 177 (93,7%) | 25 (100%) | 0,9890 |
| | Yes | 12 (5,6%) | 12 (6,3%) | 0 (0%) | |
| Main source of new syringes in the last 6 months | Other | 74 (34,6%) | 57 (40,6%) | 14 (56,0%) | 0,0146 |
| | NSP [3] | 140 (66,4%) | 129 (69,4%) | 11 (44,0%) | |
| Currently on ART | Yes | 81 (73,6%) | 76 (73,8%) | 5 (71,4%) | 0,8910 |
| | No | 29 (26,4%) | 27 (26,2%) | 2 (28,6%) | |
| **Network size** | | | | | |
| Injecting drug users | < = 10 | 147 (68,7%) | 129 68,3%) | 18 (72,0%) | 0,6924 |
| | > 10 | 67 (31,3%) | 60 (31,7%) | 7 (28,0%) | |
| Non-injecting drug users | > 3 | 51 (23,8%) | 44 (23,3%) | 7 (28,0%) | 0,6034 |
| | < = 3 | 163 (76,7%) | 145 (76,7%) | 18 (72,0%) | |
| **External norms** | | | | | |
| Any friends assisted injection initiation in the last 6 months | No | 66 (52,0%) | 56 (51,4%) | 10 (55,6%) | 0,7425 |
| | Yes | 61 (48,0%) | 53 (48,6%) | 8 (44,4%) | |
| **Initiation of others L6M: helping and promoting behaviours** | | | | | |
| Has been asked to assist with a 1st injection | No | 180 (84,1%) | 159 (84,1%) | 21 (84,0%) | 0,9870 |
| | Yes | 34 (15,9%) | 30 (15,9%) | 4 (16,0%) | |
| . . . for how many | Mean (SD) | 2,44 (3,01) | 2,63 (3,15) | 2,60 (3,17) | 0,3522 |
| | Min—Max | 1–15 | 1–15 | 1–15 | |
| Has talked positively | No | 206 (96,3%) | 182 (96,3%) | 24 (96,0%) | 0,9415 |
| | Yes | 8 (3,7%) | 7 (3,7%) | 1 (4,0%) | |
| . . . to how many | Mean (SD) | 1,75 (0,71) | 1,85 (0,69) | 1,86 (0,69) | 0,2320 |
| | Min—Max | 1–3 | 1–3 | 1–3 | |
| Has injected in front of a non-injector | No | 180 (84,1%) | 159 (84,1%) | 21 (84,0%) | 0,9870 |
| | Yes | 34 (15,9%) | 30 (15,9%) | 4 (16,0%) | |
| . . . how many | Mean (SD) | 1,94 (0,89) | 1,87 (0,82) | 1,87 (0,82) | 0,2793 |
| | Min—Max | 1–4 | 1–4 | 1–4 | |
| Has offered to give a 1st injection | No | 209 (97,7%) | 184 (97,4%) | 25 (100%) | 0,9892 |
| | Yes | 5 (2,3%) | 5 (2,6%) | 0 (0%) | |
| . . . to how many | Mean (SD) | 1,40 (0,89) | 1,4 (0,89) | - | - |
| | Min—Max | 1–3 | 1–3 | - | |
| Has assisted with a 1st injection | No | 201 (93,9%) | 176 (93,6%) | 23 (95,8%) | 0,6725 |
| | Yes | 13 (6,1%) | 12 (6,4%) | 1 (4,2%) | |

[1] Receptive sharing—getting used syringes or needles to use for own injections

[2] Distributive sharing—giving, lending, renting, or selling syringes or needles, that the individual has already used, to someone else to inject with

[3] NSP—Needle and syringe program

Table 2 presents characteristics of the retained PWID for baseline and six months after the intervention. Among the participants retained, in the six months prior to the baseline interview, 3.7% reported that they had spoken positively about injection to a non-injector, 15.9% had injected in front of a non-injector, and very few (n = 3) reported offering to give a non-injector a first injection. One sixth (15.9%) had been asked to assist with a first injection. Only a small minority (n = 12; 6.4%) reported that they had assisted someone with a first injection. At the post-intervention follow up, the proportion of those who within six months had injected in front of non-PWID declined from 15.9% to 8.5% (p = 0,0216), and reporting assisting with a first injection from 6.4% to 1.6% (p = 0,0162).

There were some changes in the drug use among study participants over the follow up. Among current injectors retained in the study, 33 (17.5%) reported not injecting drugs at the follow up (none of them were on methadone treatment). Injecting drugs other than fentanyl (i.e. stimulants, mainly amphetamine) was increased (p = 0,0104), as was using non-injection drugs (p = 0,0265). The proportion of friends assisted injection initiation in the last 6 months reported by the participants was lower at the follow-up (48,6%) than that reported for the period preceding study participation (30,6%, p = 0,0446). In comparison to baseline, a lower proportion of PWID reported distributive sharing at the follow-up (18,6 vs 10,4%, p = 0,0108).

## Changes in injection promoting behaviors and assisting with first injections

Table 3 presents the frequencies and percentages of the sample reporting "assisting" and "promoting" behaviours at baseline and six-month follow-up. Number of PWID ceasing assisting and promoting was larger than number of PWID starting with these behaviours (injecting in front of non-PWID: 30/189 vs 16/189, McNemar test p = 0,0216; assisting with 1st injection: 12/188 vs 2/188, McNemar test p = 0,0162; data was missing for 1 person on the assisting variable). As shown in Table 3, comparing the "yes/no" column with the "no/yes" column (i.e., those dropping a behaviour compared with those taking up a behaviour), fewer participants engaged in all five behaviours of interest at follow-up compared with baseline. The reductions for injecting in front of non-PWID and assisting with first injections were statistically significant. Of those who had injected in front of non-PWID at baseline, 76.7% (23/30) reported not doing so at six-month follow-up. Of those who reported assisting with first injections at baseline, 100% (12/12) reporting not doing so at follow-up.

## Discussion

Over the past two decades, the North American countries have seen a dramatic increase and Europe a modest increase in the medical and non-medical use (misuse) of prescription opioids and related fatalities [25, 26]. The US opioid epidemic has led to rising intravenous drug use and has created new public health epidemics of hepatitis C and deadly bacterial infections [27, 28]. Stemming transitions to injection drug use is therefore an important public health goal. Our study contributes to the limited knowledge base on strategies to prevent injecting initiation.

Our results provide support for the study hypotheses that after receiving the intervention, there would be a reduction in the number of participants who report "assisting" with first injection, and "promoting" injection by injecting in front of non-PWID from baseline to follow-up. While assisting in the last 6 months with a first injection was rarely reported among our PWID at baseline, we saw a significant reduction at follow-up. From baseline to follow-up, there was a significant decline in the most common "promoting" behaviour reported at baseline (i.e. in one-quarter of the sample): injecting in front of non-PWID. We did not see

**Table 2. Characteristics of participants at baseline, and 6 months after the intervention, people who inject drugs, Tallinn, Estonia 2018–2019.**

| Variable | Categories | Baseline n (%) | Follow-up n (%) | Baseline vs follow-up, p-value |
|---|---|---|---|---|
| **Socio-demographic characteristics** | | | | |
| Age | > 30 | 145 (76,7%) | 145 (76,7%) | |
| | < = 30 | 44 (23,3%) | 44 (23,3%) | |
| Gender | Female | 61 (32,2%) | 61 (32,2%) | |
| | Male | 128 (67,7%) | 128 (67,7%) | |
| Education | <10 years | 95 (50,3%) | 95 (50,3%) | |
| | > = 10 years | 94 (49,7%) | 86 (45,5%) | |
| Employment | Not employed | 89 (47,1%) | 79 (41,8%) | 0,2120 |
| | Employed | 100 (52,9%) | 110 (58,2%) | |
| Place of residence | Unstable housing | 32 (16,9%) | 43 (22,8%) | 0,0455 |
| | Stable housing | 157 (83,1%) | 146 (77,2%) | |
| **Injection drug use (in the last 6 months)** | | | | |
| Length of injection drug use (lifetime) | < = 5 years | 8 (4,2%) | 8 (4,2%) | |
| | > 5 years | 181 (95,8%) | 181 (95,8%) | |
| Main drug injected | Other | 118 (62,4%) | 135 (71,4%) | 0,0104 |
| | Fentanyl | 71 (37,6%) | 54 (28,6%) | |
| Does not injected drugs | | na | 33 (17,5%) | |
| Any non-injection drug use | No | 11 (5,8%) | 2 (1,1%) | 0,0265 |
| | Yes | 178 (94,2%) | 187 (98,9%) | |
| Injecting daily (in the last 4 weeks) | Daily | 37 (21,6%) | 24 (20,5%) | 0,6892 |
| | Less frequent | 134 (78,4%) | 93 (79,5%) | |
| Receptive sharing | No | 156 (83,0%) | 157 (85,8%) | 0,5959 |
| | Yes | 32 (17,0%) | 26 (14,2%) | |
| Distributive sharing | No | 153 (81,4%) | 164 (89,6%) | 0,0108 |
| | Yes | 35 (18,6%) | 19 (10,4%) | |
| **Sexual behaviour (in the last 6 months)** | | | | |
| Any sex partners | Yes | 163 (86,2%) | 149 (79,7%) | 0,0093 |
| | No | 26 (13,8%) | 38 (20,3%) | |
| ..Any unprotected sex | Yes | 136 (83,4%) | 120 (79,5%) | 0,8383 |
| | No | 27 (16,6%) | 31 (20,5%) | |
| **HIV infection** | | | | |
| HIV seropositivity | Pos | 103 (54,5%) | 104 (55,0%) | > 0,95 |
| | Neg | 86 (44,5%) | 85 (45,0%) | |
| **Treatment and harm reduction services utilization** | | | | |
| Currently on methadone | No | 177 (93,7%) | 187 (98,9%) | 0,0094 |
| | Yes | 12 (6,3%) | 2 (1,1%) | |
| Main source of new syringes in the last 6 months | Other | 57 (40,6%) | 40 (26,1%) | 0,5218 |
| | NSP | 129 (69,4%) | 113 (73,9%) | |
| Currently on ART | Yes | 76 (73,8%) | 79 (76,0%) | 0,6464 |
| | No | 27 (26,2%) | 25 (24,0%) | |
| **Network size** | | | | |
| Injecting drug users | < = 10 | 129 (68,3%) | 133 (70,4%) | 0,6025 |
| | > 10 | 60 (31,7%) | 56 (29,6%) | |
| Non-injecting drug users | > 3 | 44 (23,3%) | 34 (18,0%) | 0,2120 |
| | < = 3 | 145 (76,7%) | 155 (82%) | |
| **External norms** | | | | |
| Any friends assisted injection initiation in the last 6 months | No | 56 (51,4%) | 77 (69,4%) | 0,0446 |

*(Continued)*

**Table 2.** (*Continued*)

| Variable | Categories | Baseline n (%) | Follow-up n (%) | Baseline vs follow-up, p-value |
|---|---|---|---|---|
| | Yes | 53 (48,6%) | 34 (30,6%) | |
| **Initiation of others L6M: helping and promoting behaviours** | | | | |
| Has been asked to assist with a 1st injection | No | 159 (84,1%) | 169 (89,4%) | 0,1116 |
| | Yes | 30 (15,9%) | 20 (10,6%) | |
| . . . for how many | Mean (SD) | 2,63 (3,15) | 1,75 (1,12) | 0,6160 |
| | Min—Max | 1–15 | 1–5 | |
| Has talked positively | No | 182 (96,3%) | 185 (97,9%) | 0,5050 |
| | Yes | 7 (3,7%) | 4 (2,1%) | |
| . . . to how many | Mean (SD) | 1,85 (0,69) | 1 (0) | 0,3173 |
| | Min—Max | 1–3 | 1–1 | |
| Has injected in front of a non-injector | No | 159 (84,1%) | 173 (91,5%) | 0,0216 |
| | Yes | 30 (15,9%) | 16 (8,5%) | |
| . . . how many | Mean (SD) | 1,87 (0,82) | 2,81 (1,83) | 0,2685 |
| | Min—Max | 1–4 | 1–7 | |
| Has offered to give a 1st injection | No | 184 (97,4%) | 189 (100%) | 0,0736 |
| | Yes | 5 (2,6%) | 0 (0%) | |
| . . . to how many | Mean (SD) | 1,4 (0,89) | - | - |
| | Min—Max | 1–3 | - | |
| Has assisted with a 1st injection | No | 176 (93,6%) | 186 (98,4%) | 0,0162 |
| | Yes | 12 (6,4%) | 3 (1,6%) | |

significant changes in the promoting behaviours that were rare (i.e. talking positively about injection, offering to give first injection to a non-PWID). Further, we saw pre-post reductions in participants' own drug injecting. Among current injectors retained in the study, close to one fifth (17.5%) reported not injecting drugs at the follow up. As a potential positive secondary effect of the intervention, this finding warrants careful attention and evaluation in further studies. Our pre-post design for evaluating enhanced BtCag endorses this as a distinctly promising intervention.

Results reported here are generally consistent with previous data reported data reported from trials of Break the Cycle [7], and Change the Cycle [9]. Our intervention was adapted from the Break the Cycle intervention (Hunt et al [7]). In response to the recognised need the

**Table 3. Changes in injection promoting and initiation assisting behaviours from baseline to follow-up among people who inject drugs, Tallinn, Estonia 2018–2019.**

| | Behaviour changes—baseline, follow-up | | | | Mcnemar test p-value |
|---|---|---|---|---|---|
| | n = 189* | | | | |
| | No, No | Yes, No | No, Yes | Yes, Yes | |
| Has been asked to assist with a 1st injection | 148 | **21** | **11** | 9 | 0,1116 |
| *Promotion behaviour* | | | | | |
| Has talked positively | 179 | **6** | **3** | 1 | 0,5050 |
| Has offered to give a 1st injection | 184 | **5** | **0** | 0 | 0,0736 |
| Has injected in front of a non-injector | 150 | **23** | **9** | 7 | 0,0216 |
| *Assisting behaviour* | | | | | |
| Has assisted with a 1st injection | 174 | **12** | **2** | 0 | 0,0162 |

* Data was missing for 1 person on the assisting variable

module on safe injection practices was included in the intervention tested in Tallinn and new modules on spreading norms to other PWID of refraining from assisting with first injections and on overdose prevention (including information on naloxone) were added.

There are great differences in injecting drug use epidemics throughout the world, including the size and stage of the epidemic the drugs being injected, the health services available to persons who use drugs and the characteristics of the persons injecting drugs. All of these factors could potentially influence the effectiveness of Break the Cycle type interventions to reduce initiation into injecting drug use. There was a dramatic change in injecting drug use in 2017 in Estonia. The clandestine laboratory that was the dominant source for fentanyl was shut down, leading to a severe shortage of fentanyl. The changes included increases in amphetamine injecting and increases in the use of "novel psychoactive substances" (NSPs), and discontinuation of medication-assisted (methadone) treatment [29]. We previously conducted a trial of Break the Cycle in 2016–2017 prior to the fentanyl shortage in Tallinn [10]. The same RDS recruitment and follow-up methods, and intervention were used in both studies. Comparison of the results for these two trials highlights a very consistent effect of the intervention (in 2016–2017 the percentages assisting with first injections declined from 4.7 to 1.3%, 73% reduction; in the current study 83% reduction) and in a way this accentuates the robustness of our intervention effect within the target population.

The results presented here should be interpreted acknowledging the limitations of the study. This study had a modest sample size, which influenced its statistical power. Nevertheless, important differences over time were observed. Obtaining probability samples of PWID populations is challenging due to the hidden nature of this group, their stigmatised behaviours and the absence of a sampling frame. Although, RDS surveys have demonstrated the ability to reach hidden population sub-groups, the representativeness of our samples cannot be verified. We achieved a modest rate of attrition over time (12%), and there is the potential for participant loss to follow-up to have biased results. Attrition was, however, not associated with promoting or helping behaviours at baseline. Another limitation is relying on participant self-report. Social desirability responses are a possible factor in our results.

We measured outcomes six months after the intervention and we do not know if and for how long any behavioural changes were sustained beyond this timeframe. Quasi-experimental design was chosen over the randomised control design to assess the effect of the intervention. We are fully aware of the strengths of randomisation but also considered possibility of contamination/diffusion of the intervention (as study sampling relied on social networks and our intervention included a component of talking with peers to discourage assisting with first injections) and ethical aspects (refraining from providing a potentially needed intervention from part of the study participants) to be important enough to consider. A stepped wedge cluster randomised trial would be important for future assessments of the BtCag intervention. If a stepped wedge randomized trial shows an effect size similar to the effect size in the pre- versus post trials, then the intervention should be scaled up to study a community-wide effect.

In conclusion, in the context of established and emerging injection drug use epidemics, there is a need to prevent initiation into injection drug use. Within the limits of our study, the enhanced BtCag intervention adapted for the use in the setting of high prevalence of HIV and moderate prevalence of injection assisting tested proved to be effective and safe.

## Supporting information

**S1 Checklist. TREND statement checklist.**
(PDF)

**S1 Protocol.**
(PDF)

**S2 Protocol.**
(PDF)

**S1 Questionnaire.**
(DOCX)

## Author Contributions

**Conceptualization:** Anneli Uusküla, Don C. Des Jarlais.

**Data curation:** Mait Raag.

**Formal analysis:** Mait Raag.

**Funding acquisition:** Anneli Uusküla, Don C. Des Jarlais.

**Investigation:** Talu Ave.

**Methodology:** Anneli Uusküla, David M. Barnes, Susan Tross, Don C. Des Jarlais.

**Resources:** Anneli Uusküla.

**Supervision:** Anneli Uusküla, Talu Ave.

**Writing – original draft:** Anneli Uusküla.

**Writing – review & editing:** Mait Raag, David M. Barnes, Susan Tross, Talu Ave, Don C. Des Jarlais.

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
