## [Decision Letter · Decision Letter 0]

26 Aug 2022

PONE-D-22-05529Adapted “Break The Cycle for Avant Garde” Intervention to Reduce Injection Assisting and Promoting Behaviours in People Who Inject Drugs in Tallinn, Estonia: A Pre- Post Trial.PLOS ONE

Dear Dr. Uusküla,

Thank you for submitting your manuscript to PLOS ONE. After careful consideration, we feel that it has merit but does not fully meet PLOS ONE’s publication criteria as it currently stands. Therefore, we invite you to submit a revised version of the manuscript that addresses the points raised during the review process.

 Your manuscript has been reviewed by two peer-reviewers and their reports are appended below. The reviewers comment that the study could be strengthened by additional detail on the study's methodology and further explanation on the experimental design. Furthermore, the reviewers have commented that the statistics used in this study could be more clearly described, justified, and reported. Could you please revise the manuscript to carefully address the concerns raised?

We look forward to receiving your revised manuscript.

Kind regards,

Maria Elisabeth Johanna Zalm, Ph.D

Editorial Office

PLOS ONE

Journal Requirements:

AU - grant # IUT34-17 from the Estonian Ministry of Education and Research, Estonia

DDJ - grant 1DP1DA039542 from the National Institutes of Health, USA

This work was supported by grant 1DP1DA039542 from the National Institutes of Health, USA, and by grant # IUT34-17 from the Estonian Ministry of Education and Research.

However, funding information should not appear in the Acknowledgments section or other areas of your manuscript. We will only publish funding information present in the Funding Statement section of the online submission form. 

AU - grant # IUT34-17 from the Estonian Ministry of Education and Research, Estonia

DDJ - grant 1DP1DA039542 from the National Institutes of Health, USA

None

6. We note that you have indicated that data from this study are available upon request. PLOS only allows data to be available upon request if there are legal or ethical restrictions on sharing data publicly. For more information on unacceptable data access restrictions, please see http://journals.plos.org/plosone/s/data-availability#loc-unacceptable-data-access-restrictions. 

8. We note that the original protocol file you uploaded contains a confidentiality notice indicating that the protocol may not be shared publicly or be published. Please note, however, that the PLOS Editorial Policy requires that the original protocol be published alongside your manuscript in the event of acceptance. Please note that should your paper be accepted, all content including the protocol will be published under the Creative Commons Attribution (CC BY) 4.0 license, which means that it will be freely available online, and any third party is permitted to access, download, copy, distribute, and use these materials in any way, even commercially, with proper attribution.

Therefore, we ask that you please seek permission from the study sponsor or body imposing the restriction on sharing this document to publish this protocol under CC BY 4.0 if your work is accepted. We kindly ask that you upload a formal statement signed by an institutional representative clarifying whether you will be able to comply with this policy. Additionally, please upload a clean copy of the protocol with the confidentiality notice (and any copyrighted institutional logos or signatures) removed.

Reviewers' comments:

Reviewer's Responses to Questions

**Comments to the Author**

1. Is the manuscript technically sound, and do the data support the conclusions?

Reviewer #1: Partly

Reviewer #2: Yes

2. Has the statistical analysis been performed appropriately and rigorously? 

Reviewer #1: No

Reviewer #2: I Don't Know

3. Have the authors made all data underlying the findings in their manuscript fully available?

Reviewer #1: Yes

Reviewer #2: Yes

4. Is the manuscript presented in an intelligible fashion and written in standard English?

Reviewer #1: Yes

Reviewer #2: Yes

5. Review Comments to the Author

Reviewer #1: The objective of this pre-post study is to assess changes to injection initiation assisting and promoting behaviours in participating PWID, during the six months following an adapted BTC session. The study was approved by the respective Ethics/IRB board and has a valid NCT number. The study objectives sound interesting. I have the following questions.

1. A pre-post study is like a quasi-experimental design. The manuscript is submitted as a Clinical Trial. What is the justification here?

2. Sample size/power shoiuld be generated using the "primary response variable". It was never clearly mentioned what's the primary response.

3. The subsection on Outcome variables do not mention the nature of the responses; are they continuous, discrete, proportions?

3. The statistical methods are straightforward; during the sign test application, what was the null hypothesis under consideration? Write clearly.

4. A significant number of covariates are available; it is not clear why a formal regression analysis was not conducted. The sample size looks decent enough to conduct a regression analysis.

5. Any statement of significance/non-significance in the Results section should be accompanied by a p-value. This needs to be properly checked throughout.

6. The Conclusions section should allude to future studies (with larger sample sizes), often from multicenter scenarios combining other populations, to properly validate the effect of the BEATVIC group sessions.

Reviewer #2: Preventing transitions into injection drug use is likely to have significant benefits including reductions in bacterial and infectious diseases and risk for these diseases. The study, using a pre- post-design examined the impact of the Break the Cycle intervention. While this intervention has been around since the late 1990s it has not been subject to rigorous evaluation. The proposed study improves on prior studies by using a larger sample size. I have noted a number of concerns below. One deserves more attention. Specifically, the authors need to better example how they conducted the comparison between those who were followed and those who were not. I wonder if the authors could control for the few variables that were found to differ in the final model of intervention effects. Lastly, given that there were recent changes in the drug use pattern in the study community that authors might want to examine if secular trends (or time) influenced the results.

Minor/specific comments

Page 3, line 50/51. The authors might consider removing the mention to Scott County, Indiana since there have been multiple HIV outbreaks in the USA. See:

Lyss, S.B., Buchacz, K., McClung, R.P., Asher, A., Oster, A.M., 2020. Responding to Outbreaks of Human Immunodeficiency Virus Among Persons Who Inject Drugs—United States, 2016–2019: Perspectives on Recent Experience and Lessons Learned. The Journal of Infectious Diseases 222(Supplement_5), S239-S249.

Page 5, line 100/101. The sentence beginning, “The same,…” seems incomplete. I think the authors are indicating that the adaption process and study protocol are described in citation 9. Please clarify.

Page 6, line 127. I think SIEMENS should be in [ ].

Page 6, line 139. I think this section should just begin with a statement about what survey instrument was used and perhaps a statement about its validity and reliability if available. The sentence suggest that the same questionnaire was used throughout the study, which is assumed. So I would recommend removing “In both years” as well.

Page 10, lines 226/227. I think the paraphenthical statement can be removed or if it stays, the authors should revise it to read “injecting with used syringes and give used syringes to others to inject with.”

Table 1. Not sure why p-values are shown for some variables but not others (age, gender, education among others).

Page 13,lines 241/244. According to Table 1, other variables were also found to differ (p<0.05) by follow-up including housing status, currently on methadone among other items. Not sure why only 3 are discussed here. My guess is that the authors conducted some type of multivariate regression analysis to identify factors independently associated with follow-up. If that is the case, the authors should add a sentence or two describing there approach here. I’m also a bit worried about the results reported. The bivariate p between followed and not followed for HIV status was p=0.95 and the percentages of people followed and not followed are pretty close. So some explanation about what they did here would be helpful. Whatever statistical approach they used to account to loss to follow-up should be described in the statistical methods section.

6. PLOS authors have the option to publish the peer review history of their article (what does this mean?). If published, this will include your full peer review and any attached files.

Reviewer #1: No

Reviewer #2: No

---

## [Author Response · Author response to Decision Letter 0]

22 Sep 2022

PONE-D-22-05529

,Manuscript “Adapted “Break The Cycle for Avant Garde” Intervention to Reduce Injection Assisting and Promoting Behaviours in People Who Inject Drugs in Tallinn, Estonia: A Pre- Post Trial” 

Please find a response to reviewers below.

We thank reviewers for the positive comments regarding our work.

Reviewer #1

The objective of this pre-post study is to assess changes to injection initiation assisting and promoting behaviours in participating PWID, during the six months following an adapted BTC session. The study was approved by the respective Ethics/IRB board and has a valid NCT number. The study objectives sound interesting. I have the following questions.

1. A pre-post study is like a quasi-experimental design. The manuscript is submitted as a Clinical Trial. What is the justification here?

Response: We used the following definition of a clinical trial - A research study in which one or more human subjects are prospectively assigned to one or more interventions (which may include placebo or other control) to evaluate the effects of those interventions on health-related biomedical or behavioral outcomes. Interventions may be medical products, such as drugs or devices; procedures; or changes to participants' behaviour. While not as powerful as randomized controlled trials, pre vs. post studies are a very common form of intervention evaluations and clinical trials. 

2. Sample size/power should be generated using the "primary response variable". It was never clearly mentioned what's the primary response.

Response: The primary response variable was “assisting with a first injection” which consisted of describing or demonstrating how to inject to a non-PWID who then injects for his/her first time in front of the participant, or actually injecting a non-PWID We no present this more clearly (see page page 7, line 142).

Sample size was generated based on the primary outcome (page 7, line 160)

3. The subsection on Outcome variables do not mention the nature of the responses; are they continuous, discrete, proportions?

Response: “Assisting with a first injection” was measured as the proportion of subjects who engaged this behaviour in the 6-month pre-intervention period compared to the proportion of subjects who engaged in this behaviour during the 6-month post intervention period. The proportion of subjects who engaged in any “injection promoting” behavior was also compared for these two time periods. (outcome variables are binary; the measure is proportion) (see Table 2).

4. The statistical methods are straightforward; during the sign test application, what was the null hypothesis under consideration? Write clearly.

Response: The null hypothesis - participation in Break the Cycle will not change the percentage of participants reporting “assisting” behaviours in the 6 months prior to intervention compared to the 6 months post intervention. This is now presented in the Statistical analysis section (page 9, lines 204-206)

4. A significant number of covariates are available; it is not clear why a formal regression analysis was not conducted. The sample size looks decent enough to conduct a regression analysis.

Response: We did have a substantial sample size, but the absolute number of subjects who changed from assisting with a first injection in the pre-intervention to not assisting in the post intervention period was only 12. There were 2 subjects who changed from not assisting pre-intervention to assisting post intervention. These numbers were highly significant by sign test and McNemar test. However, we had a very large number of potential predictor variables—28 relevant variables at baseline, 28 at follow-up, and 28 for possible change in these variables between baseline and follow. Many of these 84 variables were intercorrelated. We did not think we had a sufficient number of subjects who ceased assisting for a regression analysis 

5. Any statement of significance/non-significance in the Results section should be accompanied by a p-value. This needs to be properly checked throughout.

Response: Checked, and p-values added.

6. The Conclusions section should allude to future studies (with larger sample sizes), often from multicenter scenarios combining other populations, to properly validate the effect of the BEATVIC group sessions.

Response: We agree that future research should involve larger sample sized and multiple research sites. Ideally, the numbers of subjects at each research site would be large enough to creates a cultural change in the local PWID population towards not assisting with first injections. This is now included in the Discussion section on page ***

(Of note, we did not test BEATVIC group sessions)

Reviewer 2

Preventing transitions into injection drug use is likely to have significant benefits including reductions in bacterial and infectious diseases and risk for these diseases. The study, using a pre- post-design examined the impact of the Break the Cycle intervention. While this intervention has been around since the late 1990s it has not been subject to rigorous evaluation. The proposed study improves on prior studies by using a larger sample size. 

I have noted a number of concerns below. One deserves more attention. Specifically, the authors need to better example how they conducted the comparison between those who were followed and those who were not. I wonder if the authors could control for the few variables that were found to differ in the final model of intervention effects. Lastly, given that there were recent changes in the drug use pattern in the study community that authors might want to examine if secular trends (or time) influenced the results.

Response: The reviewer is correct that we did not adequately present the baseline and follow-up data in Table 1. The low p values in the current table 1 represent change from baseline to follow-up period for those who were followed, not differences in those who were and were not followed. 

To be clearer we now include two tables. The first table presents be baseline characteristics for all subjects, and baseline characteristics for those followed and those not followed, with statistical comparisons of those followed and those not followed. The second table presents baseline characteristics of those who were followed compared to their values. 

** In Table 1, Baseline characteristics of those retained and those lost to follow-up are compared using multivariable analysis. 

** We did not model the effect given the reasons re: Rev 5.4

** Secular trends - our follow up period was relatively short (6 months) and that neither we nor our colleagues in Tallinn (NSP program Convictus staff, other drug treatment researchers) noted any secular trends. It is also difficult to determine what type of secular trend would lead to the large reduction in the percentage of PWID who assisted with first injections over a limited time period.

Minor/specific comments

Page 3, line 50/51. The authors might consider removing the mention to Scott County, Indiana since there have been multiple HIV outbreaks in the USA. 

Response: We revised referencing as suggested and added paper by Lyss et al (2019)

Page 5, line 100/101. The sentence beginning, “The same,…” seems incomplete. I think the authors are indicating that the adaption process and study protocol are described in citation 9. Please clarify.

Response: Corrected.

Page 6, line 127. I think SIEMENS should be in [ ].

Response: Corrected

Page 6, line 139. I think this section should just begin with a statement about what survey instrument was used and perhaps a statement about its validity and reliability if available. The sentence suggest that the same questionnaire was used throughout the study, which is assumed. So I would recommend removing “In both years” as well.

Response: Corrected

Page 10, lines 226/227. I think the paraphenthical statement can be removed or if it stays, the authors should revise it to read “injecting with used syringes and give used syringes to others to inject with.”

Response: Corrected (page 10, line 234)

Table 1. Not sure why p-values are shown for some variables but not others (age, gender, education among others).

Response: In the first submission, Table 1 compared baseline and follow-up among those who retained in the study (Therefore, it’s not meaningful to compare age, gender and education of the same people.) In the revision we included an additional table to be clearer with the results presentation.

Page 13,lines 241/244. According to Table 1, other variables were also found to differ (p<0.05) by follow-up including housing status, currently on methadone among other items. Not sure why only 3 are discussed here. My guess is that the authors conducted some type of multivariate regression analysis to identify factors independently associated with follow-up. If that is the case, the authors should add a sentence or two describing there approach here. I’m also a bit worried about the results reported. The bivariate p between followed and not followed for HIV status was p=0.95 and the percentages of people followed and not followed are pretty close. So, some explanation about what they did here would be helpful. Whatever statistical approach they used to account to loss to follow-up should be described in the statistical methods section.

Response: Please see our response to the first critique (creating a separate table comparing those followed vs those not followed for the revised manuscript). Also, we added description of the statistical approach used to assess factors associated with the loss to follow-up into the statistical analysis section.

We would again like to thank the reviewers for their constructive comments on the paper. We believe that we have appropriately addressed all of the comments and that the paper has been substantially strengthened as a result. 

With kind regards

Anneli Uusküla

---

## [Decision Letter · Decision Letter 1]

18 Jan 2023

Adapted “Break The Cycle for Avant Garde” Intervention to Reduce Injection Assisting and Promoting Behaviours in People Who Inject Drugs in Tallinn, Estonia: A Pre- Post Trial.

PONE-D-22-05529R1

Dear Dr. Uusküla,

We’re pleased to inform you that your manuscript has been judged scientifically suitable for publication and will be formally accepted for publication once it meets all outstanding technical requirements.

Kind regards,

Ricky N. Bluthenthal

Guest Editor

PLOS ONE

Additional Editor Comments (optional):

The revised manuscript has been very responsive to comments from the prior review. Issues raised by reviewer 2 are minor and can be addressed during the proofing process.

Reviewers' comments:

Reviewer's Responses to Questions

**Comments to the Author**

1. If the authors have adequately addressed your comments raised in a previous round of review and you feel that this manuscript is now acceptable for publication, you may indicate that here to bypass the “Comments to the Author” section, enter your conflict of interest statement in the “Confidential to Editor” section, and submit your "Accept" recommendation.

Reviewer #1: All comments have been addressed

Reviewer #2: All comments have been addressed

2. Is the manuscript technically sound, and do the data support the conclusions?

Reviewer #1: (No Response)

Reviewer #2: Yes

3. Has the statistical analysis been performed appropriately and rigorously? 

Reviewer #1: (No Response)

Reviewer #2: Yes

4. Have the authors made all data underlying the findings in their manuscript fully available?

Reviewer #1: (No Response)

Reviewer #2: (No Response)

5. Is the manuscript presented in an intelligible fashion and written in standard English?

Reviewer #1: (No Response)

Reviewer #2: No

6. Review Comments to the Author

Reviewer #1: (No Response)

Reviewer #2: The revised manuscript looks good. I noticed a few minor issues that can be addressed quickly. See below.

Line 26, spell out PWID the first time it appears in the abstract and in the paper (line 47).

Line 79 add (BTC) after “Break The Cycle” as this abbreviation is used later in the paper. The authors can delete the BTC in line 174 since the abbreviation will have been defined earlier in the paper.

Line 157, spell out RDS the first time it appears.

Line 205, spell out “OR” and “CI” the first time they appear.

Line 230, remove Second “%”

Line 249, spell out “aOR” the first time it appears.

250, remove the second aOR in this line.

Table 2, page 15 the cell with “Any unprotected sex” has unnecessary “..” in it.

Line 299, previous the authors have used “injection drug use” and not intravenous drug use. Not sure why they changed the language in this line.

7. PLOS authors have the option to publish the peer review history of their article (what does this mean?). If published, this will include your full peer review and any attached files.

Reviewer #1: No

Reviewer #2: No

---

## [Editor Report · Acceptance letter]

22 May 2023

PONE-D-22-05529R1 

Adapted “Break The Cycle for Avant Garde” Intervention to Reduce Injection Assisting and Promoting Behaviours in People Who Inject Drugs in Tallinn, Estonia: A Pre- Post Trial. 

Dear Dr. Uusküla:

I'm pleased to inform you that your manuscript has been deemed suitable for publication in PLOS ONE. Congratulations! Your manuscript is now with our production department. 

Kind regards, 

on behalf of

Dr. Ricky N. Bluthenthal 

Guest Editor

PLOS ONE